# Dynamic Interplay Between miR-124-3p and EGF in the Regulation of Overgrowth via RNA Signaling

**DOI:** 10.3390/biom15081186

**Published:** 2025-08-18

**Authors:** Keziban Korkmaz Bayram, Arslan Bayram, Zeynep Yilmaz Sukranli, Ecmel Mehmetbeyoglu Duman, Fatma Aybuga, Esra Tufan Benli, Serpil Taheri, Yusuf Ozkul, Minoo Rassoulzadegan

**Affiliations:** 1Department of Medical Genetics, Faculty of Medicine, Yıldırım Beyazıt University, Ankara 06690, Türkiye; 2Betul-Ziya Eren Genome and Stem Cell (GENKÖK) Center, Erciyes University, Kayseri 38039, Türkiye; zeynepys@erciyes.edu.tr (Z.Y.S.); ecmel@erciyes.edu.tr (E.M.D.); fatma.aybuga@gmail.com (F.A.); esratfan@gmail.com (E.T.B.); staheri@erciyes.edu.tr (S.T.); ozkul@erciyes.edu.tr (Y.O.); minoo@erciyes.edu.tr (M.R.); 3Gene Targeting and Transgenic Models Platform, Izmir Biomedicine and Genome Center (IBG), Izmir 35340, Türkiye; 4GENTAN-Genetic Diseases Evaluation Centre, Izmir 35100, Türkiye; ruslanbayramov@erciyes.edu.tr; 5Department of Medical Genetics, Faculty of Medicine, Erciyes University, Kayseri 38280, Türkiye; 6Department of Medical Biology, Faculty of Medicine, Erciyes University, Kayseri 38280, Türkiye; 7Physiology Department, Faculty of Medicine, Erciyes University, Kayseri 38280, Türkiye; 8The National Institute of Health and Medical Research (INSERM)-Centre National de la Recherche Scientifique (CNRS), Université Côte d'Azur, Inserm, 06000 Nice, France

**Keywords:** pronuclear microinjection, overgrowth, neurospheres, growth factor

## Abstract

**Background:** Epigenetic mechanisms and RNA signalling profoundly impact body growth during the early stages of embryonic development. RNA molecules, like microRNAs, play a vital role in early embryonic development, laying the groundwork for future growth and function. miR-124-3p microinjected into mouse fertilised eggs (miR-124-3p*) exhibited a significantly overgrowth phenotype. Behavioural test results showed that miR-124-3p mice were more physically active, as indicated by total distance and movement velocity. However, the molecular mechanism leading to these phenotypic changes mediated by miR-124-3p remains a mystery. This study aimed to investigate the role of epidermal growth factor (EGF) in developing an overgrowth phenotype in miR-124-3p* mice. **Results:** In this research, we preferred to work with neurospheres (NSs) due to the challenges of handling a single embryo, as NSs exhibit similar features, especially regarding cell growth, differentiation, and capacity for self-renewal. We examined the mRNA expression levels of *Sox8*, *Sox9*, *Sox10*, *Doublecortin (Dcx)*, and *Neurod1* genes, which are linked to a tiny phenotype in knockout mice, in total embryos at E7.5 and hippocampal cells isolated from E19.5-day fetus and neurospheres aged 12 and 21 days, which were derived from these hippocampal cells through primary cell culture. These genes are significantly overexpressed in miR-124-3p* NSs, but not in the E7.5 total embryos or the hippocampus of the E19.5 fetus. **Conclusions:** These findings suggest a possible link between miR-124-3p microinjection and EGF activation, which may be associated with early neurogenesis and neuronal differentiation in embryos. This molecular shift might contribute to the development of mice exhibiting increased physical activity and enlarged body size, although these observations remain correlative and require further validation.

## 1. Introduction

Considering transgenic animals in the context of genetically modified organisms, the long-term effects of consuming products derived from these animals on human health still need to be fully understood [1]. There is concern that the presence of foreign DNA in the genomes of these genetically modified animals could pose unpredictable risks when consuming their meat or milk. This concern creates a dilemma for consumers. Although the prospect of producing overgrown transgenic animals more quickly and at a lower cost for increased meat and milk production is appealing, the uncertainties surrounding the potential health risks associated with foreign DNA prevent us from fully realising the economic benefits that these technologies might offer.

During the prenatal and postnatal periods, mammalian body size is shaped by a complex interplay of genetic and epigenetic factors. Microinjection of miR-124-3p into the pronucleus of the fertilised egg results in mice with an overgrowth phenotype both prenatally and postnatally [2]. miR-124-3p is a microRNA abundant in the brain, particularly in the hippocampus [3]. It is essential for neuronal differentiation [4], maturation [5], and synapse formation [6], and supports neuronal survival and apoptosis. Its conserved sequence and expression patterns throughout evolution may highlight its critical role in nervous system development.

In this study, due to the difficulties associated with working with a single embryo at the single-cell stage, we chose to work with NSs, which are known to exhibit similar common characteristics, particularly in cellular proliferation, differentiation, and self-renewal capacity [7]. In E7.5-day-old total embryos, hippocampal cells isolated from E19.5-days-old fetus, and NSs derived from these cells at days 12 and 21 in primary culture, we investigated the mRNA expression levels of *Sox8* [8], *Sox9* [2], *Sox10* [9], *Dcx* [10], and *Neurod1*, which are known to be associated with a small phenotype in mice when knocked out.

## 2. Material Methods

### 2.1. Animals

The study consists of 2 parts: prenatal (7.5-day-old embryos (*n* = 33) and 19.5-day-old embryos (*n* = 19)) and postnatal (*n* = 41). All animal and laboratory studies were performed at the ‘Betul-Ziya Eren Genome and Stem Cell Centre’ (GENKOK) facilities. The mice in the study were kept in a regulated environment with controlled lighting hours (from 6:00 a.m. to 6:00 p.m.), temperature (22 °C), and humidity (55%). All animal models were housed and cared for consistently, with stringent control over their food, water, temperature, lighting, and overall well-being. All methods were performed by the ARRIVE guidelines for reporting animal experiments, and the study was approved by the Erciyes University Animal Ethics Committee (16 November 2016) (16/132). The experiments were performed in isolated rooms during specific hours (10:00 a.m. to 4:00 p.m.), and the results pertain to the *Balb/c* mouse line.

### 2.2. RNA Microinjection in Fertilised Eggs

RNA microinjection into the pronucleus of fertilised eggs after spontaneous ovulation was performed by the established methods of DNA transgenesis [2,11,12]. Oligoribonucleotides were obtained from Sigma-Aldrich (Merck KGaA), St. Louis, MO, USA. The sequences used are presented in Appendix A. Plug-positive mice were sacrificed for microinjection; some were separated as controls. Fertilised eggs were microinjected with 500 ng/µL RNA oligonucleotide [2,13]. Anaesthesia was provided to surrogate females by intraperitoneal injection of 80–120 mg/kg ketamine (Pfizer Inc., New York, NY, USA) and 5–10 mg/kg xylazine (Bayer AG, Leverkusen, Germany). For studies of embryos during gestation, *miR-124-3p** embryos were re-implanted in the left uterine horn of the foster and were removed on the 7th and 19th days of pregnancy.

### 2.3. Three-Dimensional Hippocampal Cell Culture

Since the hippocampus develops in embryos between days 17 and 19 of gestation, hippocampal cells for 3D cell culture were obtained from embryos on day 19. The hippocampus was aspirated using an automatic pipette under a binocular microscope and then placed in 1 mL of Hanks’ Balanced Solution (HBSS). The same protocol was used for 3D cell culture, as described in the previous study [14]. One of the pairs had 20 ng/mL EGF and Insulin-Transferrin-Selenium-A (ITS) added to it. Hippocampal cells were cultured for 12 and 21 days. Images of the NSs were taken under a microscope (10×) on the 12th, 16th, and 21st days of the culture. The number and area of the NSs were quantified using the ImageJ program (version 1.52, National Institutes of Health, Bethesda, MD, USA; available at: https://imagej.nih.gov/ij/, accessed on 1 August 2025).

### 2.4. RNA Extraction, RT-PCR, Pre-Amplification, and qPCR

Total RNA was isolated from the samples (E7.5-day-old embryos, adult hippocampus, hippocampi from E19.5-day-old fetuses, and NSs cultured for 12 and 21 days from hippocampal cells isolated from these fetuses) by the phenol-chloroform technique using TRIzol (Cat No: 11667165001, Roche Diagnostics GmbH, Mannheim, Germany) (14). cDNA synthesis was performed from the obtained RNA samples using the EvoScript Universal cDNA Master kit (Cat. No. 07 912 439 001, Roche Diagnostics GmbH, Mannheim, Germany) in a 20 µL reaction volume containing 10× Enzyme Mix and 5× Reaction Buffer with a combination of random hexamer and oligo(dT)_18_ primers. The reverse transcription was carried out at 42 °C for 15 min, followed by enzyme inactivation at 85 °C for 5 min and a final incubation at 65 °C for 15 min. Preamplification was performed using the Pre-AMP Master Kit (Cat. No. 07 912 439 001, Roche, Germany) in a total reaction volume of 25 µL, including 5 µL cDNA, 5 µL Pre-AMP Master Mix, and a primer pool prepared from target genes diluted 1:10. Thermal cycling conditions were as follows: initial denaturation at 95 °C for 1 min, followed by 13 cycles of 95 °C for 15 s and 60 °C for 4 min, with a final hold at 4 °C (14). To measure mRNA expression levels, LightCycler 480 Probes Master mix (Cat No: 04902343001, Roche, Germany), primer/probe (Integrated DNA Technologies, Leuven, Belgium), and pre-amplified cDNA samples were pipetted into 96-well plates (Roche, Germany). IDT probe assays (Integrated DNA Technologies, Inc., Coralville, IA, USA) were run by qPCR with LightCycler 480 (Roche, Germany) (Appendix A). The data were analysed using the delta delta CT method and normalised using β-actin [14].

### 2.5. miRNA Expression by qPCR

Total RNA samples (E7.5-day-old embryos and adult hippocampus) were diluted to 10 ng/µL. These diluted RNA samples were converted into cDNA using the miScript II RT Kit (Qiagen GmbH, Hilden, Germany). cDNA samples were diluted at a 1:5 ratio and were mixed with 2× miScript SYBR Green Mix, 10× miScript Universal Primer, 10× miScript Primer Assay, and RNase-free water. qRT-PCR was used to measure the levels of miR-124-3p and miR-124-5p using the LightCycler 480 II (Roche). The PCR cycle consisted of 95 °C/10 min, followed by 45 cycles of 95 °C/10 s, 60 °C/30 s, and 72 °C/1 min [3]. The relative quantities of miRNAs were determined using the delta-delta Ct method, with the mouse U6 SnRNA serving as an endogenous control for the assay (Appendix A).

### 2.6. Behavioural Tests

Behavioural experiments were initiated with 8-week-old mice. Each mouse was subjected to a single test from 10:00 to 16:00. The testing instruments were cleaned with 70% ethanol between trials. Experiments were recorded on video and frequently analysed. Tests were assessed using the “EthoVision 9” software (Noldus, Wageningen, The Netherlands). Manual analysis of marble-burying tests was conducted by an observer blind to the group designation of the mice.

The novel object test generally involves two cognition evaluation trials based on a mouse’s spontaneous exploration behaviour to measure recognition memory. The social test evaluates cognitive function in mouse models related to central nervous system disorders by measuring overall sociability and curiosity about new social interactions. The tail suspension test measures immobility posture, indicating the abandonment of struggling and, thus, a potential sign of depression. The marble burying test frequently assesses rodent neophobia, encompassing hesitance towards novel objects, anxiety, and compulsive or repetitive behaviours. These tests employed the same setup and protocols as those used in our previous studies [11,12,15].

The holeboard test setup consists of 16 holes, each with a diameter of 2.2 cm, on the floor. The holes are round and placed inside a square or rectangular arena. The number of times the mice dip their heads into these holes for 5 min is recorded, and their emotional state, anxiety, and stress response are evaluated [16]. The decreased head-dipping behaviour of the mice is interpreted as an increased anxiety behaviour [17].

The setup for the open-field test consists of a box surrounded by four walls, with an open top and a base of 16 squares, equally separated by lines. The four squares in the middle are considered the centre, and the other squares are considered the periphery. The open-field test procedure was applied by placing cinnamon bark in the centre of the field. The mice were released from a corner and recorded for 5 min by the EthoVision system camera.

The Y-maze test used to measure spatial memory is a 3-armed maze (40 cm long, 3 cm wide, 12 cm high) symmetrically separated by 120 degrees, with a door in one of the arms. Mice are placed in one of the arms with cues around it, and the entrance of mice to each of the three arms is recorded for 5 min by the EthoVision system camera. The arms are designated A, B, and C. A sequence of consecutive entries into three different arms is defined as an ‘alternation’ (e.g., ABC, CAB, BCA), whereas repetitive entries (e.g., BAB) are not classified as alternations. The duration time in the arms, the number of entries into the arms, and the number of rotations are calculated [18].

### 2.7. Statistical Analyses

Data were analysed using IBM SPSS Statistics 22 (IBM SPSS Statistics for Windows, Version 22.0. Armonk, NY, USA: IBM Corp. Released 2013) software. Data distribution was analysed using a histogram, qq plot, and Shapiro–Wilk test. Independent *t*-test and dependent *t*-test tests were used to compare the data between the two groups. ANOVA was used to compare 3 or more groups, and Tukey’s multiple comparison tests were used for pairwise comparison in case of significance. The number of units (*n*), percentage (%), and mean and standard deviation values are summarised. A *p* < 0.05 (*p* < 0.01 for miRNA analysis) significance level was accepted. GraphPad Prism 6 (GraphPad Software, San Diego, CA, USA) software was used for the graphs.

## 3. Results

### 3.1. Recapitulation of the “Giant” Phenotype in the Balb/C Background Following Microinjection of miR-124-3p RNA into Fertilised Mouse Eggs

A total of 24 mice (10 males, 14 females) born after microinjection of a synthetic single-stranded miR-124-3p with the sequence of the mature product into the male pronucleus of fertilised *Balb/c* eggs were found to have an overgrowth phenotype both prenatally and postnatally (Figure 1A–D). Not only were the mice heavier postnatally at 7 days (Figure 1C), but their weekly weight during the 120 days also showed that miR-124-3p* mice were heavier than the control mice of the same gender (Figure 1D). Moreover, this overgrowth phenotype was not associated with the amount of food consumed. The amount of feed consumed by the mice was measured, and it was determined that control females consumed more feed than *miR-124-3p** females (*p* < 0.001), and *miR-124-3p** males consumed more feed than control males (Figure 1E). *miR-124-3p** males were found to be heavier by 24–44% and *miR-124-3p** females by 17–35% compared to controls (calculated by considering weighing data at day 28). *miR-124-3p** male mice were found to be heavier than the control group males on all days, especially on day 42 (*p* < 0.0001) and day 99 (*p* < 0.05). Although *miR-124-3p** female mice were generally heavier than control group females, especially on day 42 (*p* < 0.0001) and day 56 (*p* < 0.05), it is noteworthy that there is a hormonal fluctuation on some days (there is a non-menstrual cycle in mouse but the estrous cycle in female mice is typically 4–5 days) (Appendix A).

### 3.2. Behavioural Findings

In a novel object recognition experiment comparing interest in the novel object, it was found that both *miR-124-3p** males (*p* < 0.01) and females (*p* < 0.001) spent more time near the novel object than the control (Appendix A). The odour experiment measured time spent in the centre with and without odour; it was found that females were generally more affected by smell than males. Moreover, *miR-124-3p** females (*p* < 0.01) spent more time in the centre in the presence of odour than control females (Appendix A). In the Y-maze experiment, which measured short-term spatial memory, no difference was detected in *miR-124-3p** males compared to controls. In contrast, *miR-124-3p** females (*p* < 0.05) made more entries into the arms on days 1 and 2 (Appendix A). Although eight different behavioural tests were applied to the mice, it was recognised that some of the tests were not informative for *miR-124-3p** mice (Appendix A). All behavioural experiment results were evaluated, and it was determined that the behavioural differences between healthy males and females due to sex hormones were reversed between *miR-124-3p** males and females (Appendix A).

The novel object test measures rodents’ natural inclination to explore new objects more than familiar ones, demonstrating learning and recognition memory [19]. *miR-124-3p** male (*p* < 0.01) and *miR-124-3p** female (*p* < 0.001) mice showed more interest in the novel object in the test results (Appendix A). Males have better spatial abilities than females [20]. While stress impairs spatial memory in males, females can improve their spatial skills based on the type of stress [21]. No difference was found between control females and *miR-124-3p** females regarding the discrimination index (Appendix A). However, *miR-124-3p** males spent more time with the novel object than controls, indicating better learning and memory functions. *miR-124-3p** females performed similarly to control females in learning and memory but could not tolerate stress during behavioural experiments.

Since the open-field experiment was not informative, an odour experiment was performed by placing cinnamon sticks in the centre of the same experimental set-up. In this test, control (*p* < 0.05) and *miR-124-3p** females (*p* < 0.01) spent more time in the centre in the presence of odour (Appendix A). In humans, the number of cells in the olfactory bulbs of women is approximately 43% higher than in men, which contributes to women’s superior sense of smell compared to men [22]. In our study, miR-124-3p* females were found to be even more sensitive to odour than typical females. Interestingly, some of the miR-124-3p* females carried the cinnamon stick to the peripheral region of the open-field setup, where they felt safe.

In the Y-maze test, which can measure short-term spatial memory [23], it was found that *miR-124-3p** females had more entries than control females (*p* < 0.01) on the second day compared to the first day regarding the total number of entries to the arms (Appendix A). These findings suggest that miR-124-3p* females exhibited curiosity. *miR-124-3p** males performed better than control males on day one but worse on day two. *miR-124-3p** females were better on day two but worse than control females on day two compared to day one (Appendix A). These results suggest that *miR-124-3p** mice have higher alternation scores on day one but exhibit boredom-like behaviour on day two, like intelligent people who become bored quickly.

The marble burying test measures a rodent’s response to unfamiliar objects by observing its digging and burying behaviour [24]. While it is unclear which region in the neuronal system is associated with this behaviour, it is believed that the hippocampus and septum play a role in reducing digging behaviour [25]. Male mice with *miR-124-3p** did not differ from the control group, while female mice with *miR-124-3p** buried fewer marbles (Appendix A). We observed that some of the *miR-124-3p** female mice that buried less marbles were the overgrowth mice in the group (on day 21), went to the cinnamon smell the most in the olfactory experiment and spent the most time in the centre, and the mice that travelled the most distance and were the fastest in the novel object recognition experiment.

The holeboard test measures head dipping and defecation in mice. Some evidence suggests that animals are curious but fearful in unfamiliar environments [26]. We found that *miR-124-3p** male and female mice defecated more than controls (Appendix A). Defecation frequency in *miR-124-3p** mice indicates increased anxiety and stress sensitivity [27]. Female sex hormones affect gender differences in stress responses [21]. Female mice exhibit different behaviours based on their reproductive cycle. Generally, males exhibit fewer stress-related behaviours than females [28]. Control females in our study were similarly less stressed (fewer defecations) and less anxious than control males. However, although the stress levels of miR-124-3p males and females appear similar based on the number of defecations, miR-124-3p females exhibit greater anxiety-like behaviour.

### 3.3. miR-124-3p Microinjection Does Not Significantly Alter miR-124 Expression Levels in Early Embryos or Adult Hippocampus

To determine whether miR-124-3p microinjection altered miR-124 expression levels, miR-124-3p and miR-124-5p levels were measured prenatally (E7.5) and postnatally in the hippocampus. Both miR-124-3p and miR-124-5p expression levels showed no statistically significant difference between groups at either the E7.5 embryonic stage (Figure 2A) or in the adult hippocampus (Figure 2B). Even if miR-124-3p or 5p expression is overexpressed after pronuclear microinjection, leading to an overgrowth phenotype, this effect is neither long-term (not detected in the hippocampus in the adult) nor short-term enough to be detected in the E7.5 total embryo.

### 3.4. Neuronal Differentiation Could Be Promoted in miR-124-3p* Embryos During the Early Stage

The function of miR-124 in neuronal development is debated, with some studies showing that it promotes neurogenesis and others suggesting that it blocks the anti-REST/Ctdsp1 pathway [29]. Research has shown that miR-124 suppresses the expression of genes like *Sox9* and *Ptbp1* [30]. Our study observed a slight but non-significant reduction in the expression of genes, such as Ptbp1, Ptbp2, and Ctdsp1 in E7.5, which may lead to significant phenotypic and behavioural differences (Figure 3A).

### 3.5. The Role of miR-124-3p* in Regulating Body Growth Pathway During Early Embryonic Stage

Growth hormone-releasing hormone (Ghrh) is a hormone that stimulates the release of Growth hormone (Gh), which regulates overall body growth. Gh acts on target cells through its receptor, Ghr. Igf-1 mediates Gh’s effects and binds to Igf-1r to promote cell growth, proliferation, and survival. Igfbps regulate Igf-1 availability and activity by modulating its interaction with Igf-1r [31]. The activation of MAPK and PI3K/Akt pathways by Igf-1r regulates cellular processes, such as growth, proliferation, and survival [32]. In *miR-124-3p** E7.5-day-old embryos, mRNA expression levels of *Ghrh*, *Gh*, *Ghr*, *Igf*, *Igf-1*, and *Igfbp5* showed an upward trend, while *Igf-1r* expression exhibited a decrease. Among these, only *Igfbp1* expression was significantly increased (*p* = 0.0132) (Figure 3B).

### 3.6. miR-124-3p* Embryos Are Sensitive to Growth Factors Such as EGF During the Late Embryonic Period

Although EGF is not essential for NS formation in primary cultures, it was included in this study to assess the interaction between miR-124-3p* and growth signalling in vitro. Hippocampal cells isolated from an E19.5-day-old fetus were cultured on pHEMA-coated plates with or without EGF supplementation. NS formation began at approximately day 12 (Figure 4A), and at this time point, both the number and area of NSs were significantly increased in miR-124-3p* embryos compared to their respective controls in both EGF (+) (*p* < 0.0001 for number, *p* < 0.01 for area) and EGF (−) (*p* < 0.05) conditions (Figure 4B). Notably, EGF did not significantly affect NS formation in the control group, while miR-124-3p* embryos showed a marked increase in NS numbers in the presence of EGF. This enhancement persisted at later time points: on day 16, the number of NS remained elevated in miR-124-3p* embryos under both EGF (+) and EGF (−) conditions (*p* < 0.01) (Figure 4C), and by day 21, NS formation was significantly higher in EGF (+) miR-124-3p* cultures compared to EGF (−) controls (*p* < 0.01). In summary, miR-124-3p* microinjection promoted NS formation even in the absence of EGF, but the presence of EGF further amplified this effect, suggesting that miR-124-3p* embryos exhibit increased sensitivity to EGF signalling.

**Figure 4 biomolecules-15-01186-f004:**
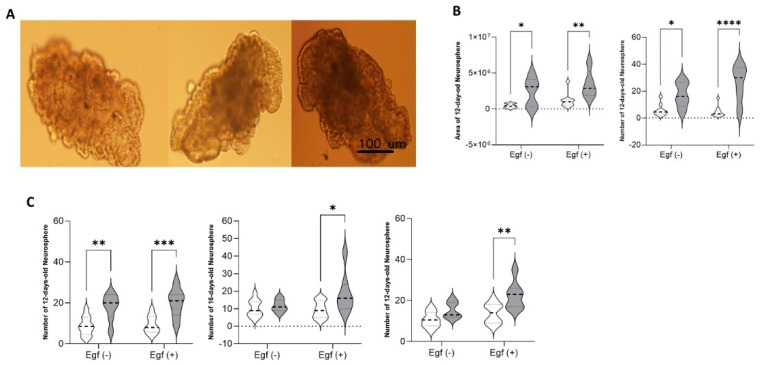
(**A**) Images of 12-, 16-, and 21-day-old NS, respectively (10×). (**B**) Area and number of 12-day-old NS; white symbol: control (*n* = 10), grey symbol: miR-124-3p* (*n* = 5). (**C**) The number of 21-day NSs that followed up 12, 16 and 21 days, respectively; white symbol: control (*n* = 11), grey symbol: miR-124-3p* (*n* = 7). *p* < 0.05 (*), *p* < 0.01 (**), *p* < 0.001 (***), *p* < 0.0001 (****).

The mRNA expression levels of *Sox8* (*p* = 0.0049), *Sox9* (*p* < 0001), and *Sox10* (*p* = 0.0220) were significantly higher in *miR-124-3p** EGF (+) 21-day-old NSs compared to controls. However, there was no change in *Neurod1* and *Dcx* mRNA expression. On the other hand, *Dcx* (*p* < 0001) and *Neurod1* (*p* = 0.0014) mRNA expression levels were significantly increased in *miR-124-3p** EGF (−) 12-day-old NSs but remained unchanged in 21-day-old NSs (Figure 5B).

## 4. Discussion

Microinjection of miR-124-3p into fertilised mouse eggs results in mice displaying a noticeable overgrowth phenotype both prenatally and postnatally (Figure 1A,B). This result indicates that a 22-nucleotide miRNA can cause a phenotypically observable change. The notable overgrowth phenotype identified by Grandjean et al. in *miR-124-3p* B6/D2* mice was observed for the first time in this study with *Balb/c* mice. This study aims to elucidate the molecular mechanism underlying the occurrence of this prominent phenotype in *miR-124-3p** mice.

Distinct genetic and epigenetic mechanisms govern body size and weight regulation in mammals during the prenatal and postnatal stages. Grandjean et al. (2) found that B6/D2 miR-124-3p* mice were approximately 30% larger. In this study, mice with *miR-124-3p** had an overgrowth phenotype prenatally (at E7.5 and E19.5) and postnatally. Compared to controls, *miR-124-3p** mice were heavier on postnatal day (PND) seven (Figure 1C) and not heavier on PND21 (Appendix A) but rapidly gained weight and became heavier in the following weeks (Appendix A). At PND7, *miR-124-3p** mice are probably heavier (Figure 1B) because they may behave more actively and consume more breast milk than the control. The presence of breast milk in the abdominal region of *miR-124-3p** at PND2 was observed before (2). At PND21, the mice stop breastfeeding, and the litter feeds themselves. However, *miR-124-3p** mice become depressed and only feed on breast milk. So, they are not heavier than the control at PND21. However, after PND21, *miR-124-3p** mice recovered and were fed more actively than the control. The findings of the behaviour tests support this; miR-124-3p* moved faster than the control (Appendix A). This study found that *Balb/c miR-124-3p** male mice were 24–44% larger, and *miR-124-3p** female mice were 17–35% larger than their controls (calculated by considering the weighing data on the 28th day). Do *miR-124-3p** mice eat more and gain more weight? To find out, we weighed the mice and their food daily. As shown in Figure 1E, there was no statistically significant difference in food intake between *miR-124-3p* males and control males. In contrast, *miR-124-3p* females consumed significantly less food than control females (*p* < 0.01).s ate less than control females (Figure 1E). This suggests that daily weighing may have caused stress in the female mice, preventing them from eating.

Epigenetically modified through microinjection of miR-124-3p, known for its expression in the hippocampus, were subjected to behavioural tests. These tests indicate that alterations in mRNA expressions may lead to the disappearance of the existing differences between males and females regarding behaviours, such as stress and anxiety in *miR-124-3p** mice (Appendix A).

PHEMA-coated plates create a non-adhesive surface that encourages the growth of NS cells (14), unlike 2D culture, where neurons tend to become glial cells due to surface adhesion [33]. NSs provide a more natural three-dimensional environment for studying neural stem cells [7]. Although they can differentiate into various cell types, they are less complex than early-stage embryos. While both models have potential for studying neurodevelopment and embryogenesis, early embryos possess a more significant cellular potential. A discussion of the findings on mRNA expression profiles in the total embryos at E7.5 days, E19.5-day-old fetuses, and hippocampal cells derived from these fetuses cultured in 3D with PHEMA for 12 and 21 days using EGF is provided below.

SoxE genes (*Sox8*, *Sox9*, and *Sox10*) play a crucial role in developing the nervous system and regulating the development of neurons, glial cells, and other necessary cell types [34]. Homozygous *Sox8* knockout mice were 30% smaller and 10% shorter than the control group after being weighed for 100 days [8]. *Sox8* mRNA expression was undetectable in E2.5 embryos but increased towards adulthood in *miR-124-3p** [2]. Our study found similarly lower *Sox8* mRNA expression in *miR-124-3p** E7.5 embryos compared to the control. (Figure 3A). However, a slightly increased *Sox8* mRNA expression was observed in *the miR-124-3p** E19.5 fetus (Figure 5A). *Sox8* mRNA expression increased in NSs with EGF (+), particularly at 12 and 21 days (*p* = 0.0028). No changes were observed in EGF (−) (Figure 5B). Sox8 may play a role in maintaining accelerated growth until adulthood.

*Sox9* mRNA expression decreased as embryogenesis progressed in *miR-124-3p* B6/D2* embryos [2]. Since heterozygous *Sox9* knockout mice died after birth, it was observed that when *Sox9* expression was reduced by siRNA transfection in the zygote, embryos were tiny, development was abnormal, and embryos died at E10.5. Thus, they stated that Sox9 is required during early development but could not detect a specific effect on cell proliferation and growth control [2]. Injecting *Sox9* cDNA downstream of the CMV1 promoter resulted in larger *miR-124-3p** E7.5-day embryos with normal morphology, but development was arrested at E11.5 days. In our study, *Sox9* mRNA expression was lower in E7.5 compared to controls (Figure 3A), while it was at the same level in E19.5-day-old fetuses (Figure 5A). However, no change was observed in *Sox9* mRNA expression level in the EGF (−) in NSs. Interestingly, in the EGF (+), *Sox9* mRNA expression level was unchanged in miR124-3p* 12-day-old NSs compared to controls, whereas *miR-124-3p** was overexpressed in 21-day NSs compared to both 12 (*p* < 0.001) and 21 (*p* < 0.0001) day control NSs (Figure 5B). Given the similarities between NSs and single-cell embryos, it can be inferred that *Sox9* may be required for the overgrowth phenotype during early development but may not have a specific effect later in the embryo. In addition, *Sox10* mRNA expression was found to be decreased in *miR-124-3p** E7.5 (Figure 3A) and E19.5 hippocampus (Figure 5A) in this study. Otherwise, *Sox10* mRNA expression increased in *miR-124-3p** 21-day-old NSs compared to controls (*p* = 0.220) and *miR-124-3p** 12-day-old NSs (*p* = 0.315) (Figure 5B). Considering the similarities between NSs and single-cell embryos, it can be inferred that *Sox10* is overexpressed in very early-stage embryos, especially in the presence of EGF. As determined in behavioural experiments, it contributes to neuronal development and makes them more active.

Dcx is a neuron-expressed protein that regulates microtubules to direct neuron migration in the nervous system during development [34]. *Dcx* knockout mice had normal growth until weaning, but lower body weight at four months [10]. *Dcx* knockout mice had slower growth rates and weighed 3–7% less than control mice in a more extensive cohort study [35]. In this study, *Dcx* mRNA expressions were at the same level as controls in *miR-124-3p** E7.5 and *miR-124-3p** E19.5 hippocampus (Figure 3A and Figure 5A). Similarly, in the EGF (−) group, miR-124-3p* *Dcx* mRNA expression was at the same levels as those of the controls in 12- and 21-day-old NSs. EGF (−), *Dcx* mRNA expression was deficient in *miR-124-3p** 21-day NSs and at the same level as the control, but *miR-124-3p** was significantly increased in the 12-day NSs (*p* < 0.0001) (Figure 5B). Ninomiya et al. found that EGF infusion in ischemic mice enhanced the proliferation of cells but inhibited their differentiation into neuroblasts [36]. Similarly, in this study, *Dcx* mRNA expression in *miR-124-3p** mice increased in the absence of EGF but decreased to control levels in its presence. Prolonged NS culture time could eliminate the neurogenesis capacity, and EGF may have negatively affected *Dcx* mRNA expression.

Neurod1, a neurogenic factor and proneuronal marker, was shown to be among the targets of miR124-3p by decreased gene expression after miR-124 microinjection into frog blastomeres [37]. In this study, *Neurod1* mRNA expression was increased in *miR-124-3p** E7.5 and E19.5 hippocampus (Figure 3A and Figure 5A). *Neurod1* mRNA expression in *miR-124-3p** 21-day-culture NS was found to be at the same level as controls and significantly lower in *miR-124-3p** 12-day-culture NS compared to controls in the presence (*p* = 0.0014) and absence (*p* = 0.0002) of EGF (Figure 5B). Following NS formation, *Neurod1* mRNA expression increases in *miR-124-3p** mice, independently of the presence of EGF. However, with prolonged culture time, its expression declines to control levels. It indicates that prolonged NS culture almost eliminates the capacity of NSs for neuronal differentiation. When the culture period is prolonged to 21 days in miR-124-3p* NSs, EGF overexpresses transcription factors, such as Sox8, Sox9, and Sox10; however, it decreases the mRNA expression of the neurogenesis marker *Dcx* and the neuronal differentiation marker *Neurod1*. These results demonstrate that EGF may have suppressed neurogenesis and neuronal differentiation in miR-124-3p* NSs with prolonged culture time. A similar mechanism could have been activated immediately after the microinjection of miR-124-3p into a fertilised egg. Following microinjection of miR-124-3p, EGF may have been activated, suppressing miR-124-3p expression and leading to early neurogenesis and neuronal differentiation in the embryos, resulting in cognitively active and overgrowth phenotypic mice.

The conservation of the sequence and expression pattern of miR-124 during the evolutionary process reveals that it is critically important in CNS development [38]. Overexpressing miR-124 through transfection in nerve progenitors of chick neuronal tubes resulted in decreased expression of small C-terminal domain phosphatase 1 (*Ctdsp1*). The 3′ UTR region of Ctdsp1 mRNA contains regions compatible with the evolutionarily conserved miR-124 sequence. Thus, Visvanathan et al. [29] suggested that miR-124 facilitates neurogenesis at least partially by blocking the neuronal anti-REST/Ctdsp1 pathway. The formation and development of neuritis increased in cell lines (HeLa and N2A) with decreased polypyrimidine tract-binding protein (*Ptbp*) expression. Therefore, it is stated that some splicing events regulated by *Ptbp* may directly lead to the observation of neuronal phenotypes. They also reported that miR-124 targeted *Ptbp1* by working as an antagonist to the REST pathway and initiated a neuron-specific splicing mechanism [39]. Mokabber et al. [30] state that the mRNA expression of *Sox9* and *Ptbp1* increased when miR-124 expression decreased due to mimicking miR-124 transfection in hair follicle stem cells. It suggests that miR-124 suppresses Sox9 and Ptbp1, which in turn increases Ptbp2 and supports neuron-specific formation [40]. In this study, *Ptbp1*, *Ptbp2*, and *Ctdsp1* mRNA expression levels in miR-124-3p* E7.5 embryos showed a slight downward trend compared to controls; however, these differences were not statistically significant (Figure 3A). These genes are known targets of miR-124-3p and are involved in RNA processing and splicing. Although our data do not reveal substantial changes, future studies may further investigate whether *miR-124-3p* influences early developmental processes—such as alternative splicing—or contributes to downstream phenotypes via post-transcriptional regulation.

All these findings suggest that mice with an overgrowth phenotype in both prenatal and postnatal periods due to miR-124-3p microinjections are more sensitive to a growth factor such as EGF. It primarily affects epithelial cells, whereas Gh affects general growth and development in various tissues and organs. Nevertheless, they have similar properties, especially in the early embryonic period. Both EGF and Gh play essential roles in regulating cellular processes and development during early embryogenesis. While their functions do not overlap entirely, there are some similarities in their effects during this stage. EGF and Gh are growth factors that promote cell division and proliferation in different cell types, contributing to overall embryonic growth. They also play crucial roles in the development and formation of organs during early embryogenesis. By regulating gene expression, they contribute to the complex orchestration of early embryonic processes and are involved in the development and function of the trophoblast [41].

The expression of miR-124-3p and miR-124-5p was measured in *miR-124-3p** mice during prenatal (E7.5-total embryo) and postnatal (8-week-old hippocampus) periods, but no difference was found compared to the control; the limitation of this study is that it was not measured in single-cell stage embryos. In addition, most embryos with *miR-124-3p** were found in twin formations within the uterus of the foster dams [2]. In this study, the *Dmrta1* mRNA expression level decreased in *miR-124-3p** E7.5 compared to controls (Figure 3A). *Dmrta1* knockout mice showed no anatomical deficiency, were viable and productive, males exhibited mating behaviour with other males, and females had polypollicular ovaries (carrying more than one egg) [42]. Our findings may explain frequent twin formations in *miR-124-3p** embryos.

The body growth pathway begins with GHRH stimulating the release of GH, which acts on target tissues through the GHR receptor [43]. Gh also induces the production of Igf-1, which binds to Igf-1r and activates downstream signalling pathways. Igfbp1 and Igfbp5 regulate the availability and activity of Igf-1. The overall coordination of these factors and their signalling pathways plays a vital role in regulating body growth, especially during the development and growth phases [44]. Grandjean et al. [2] showed that *miR-124-3p** embryos did not overexpress *Gh*, *Igf1*, *Igf2*, and their receptors at the early embryonic stage. In contrast, mRNA expressions of *Ghrh*, *Gh*, *Ghr*, *Igf*, *Igf-1*, *Igfbp5*, and *Igfbp1* (*p* = 0.0132) increased, but *Igf-1r* decreased in *miR-124-3p** E7.5-day-old embryos in this study (Figure 5A).

Due to the difficulties of working with single-cell embryos, the NS was thought to have similar common characteristics and was therefore preferred in this study. The ability of EGF to suppress miR-124 expression through the mitogen-activated protein kinase/extracellular-signal-regulated kinase (MEK) and PI3K signalling pathways [37,45] allowed the following inference to be made from the results obtained in this study: EGF is probably activated by miR-124 microinjection and suppresses miR-124-3p expression. Thus, *Sox8*, *Sox9*, *Sox10*, and *Dcx*, which, when overexpressed, can lead to an overgrowth phenotype in mice, could be downregulated in single-cell embryos, as we found in the NSs of *miR-124-3p** mice too. Just as *Gh* mRNA expression increases in E7.5, EGF, another growth factor, probably increases in the early embryonic period, especially following the microinjection of miR-124-3p to the single-cell stage embryo. Therefore, increased expression of these genes following miR-124-3p microinjection may lead to the overgrowth phenotype.

A significant limitation of this study is the lack of protein-level validation, due to the extremely low input material obtained from single embryos and fetal hippocampi. Therefore, gene expression analyses were limited to probe-based qPCR, which was chosen for its superior sensitivity and specificity in low-input samples. While this method ensures reliable detection of target transcripts, it offers no direct information on the protein-level consequences.

## 5. Conclusions

Microinjection of miR-124-3p induces an overgrowth phenotype and results in a very slight downregulation of miR-124-3p expression levels. *miR-124-3p** NSs are more sensitive to a growth factor such as EGF. Since miR-124-3p does its job and degrades after microinjection, it is essential because it may produce overgrowth phenotype animals whose meat can be consumed without foreign DNA.

## Figures and Tables

**Figure 1 biomolecules-15-01186-f001:**
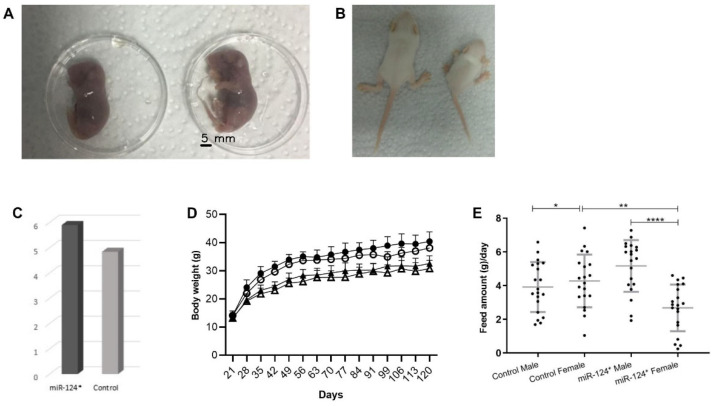
(**A**) Prenatal stage (E19.5 days-old fetus); control is on the left, miR-124-3p* is on the right. Postnatal growth: (**B**) Eleven-day-old control (right), miR-124-3p* (left) pups. (**C**) Seven-day-old body weights of the miR-124-3p* (*n* = 18) and control (*n* = 25). (**D**) miR-124-3p* (*n* = 24; 10 males, 14 females) and control (*n* = 17; eight males, nine females) mice were weighed weekly for 120 days (black symbols: miR-124-3p*, white symbols: controls, triangles: females, circles: males). (**E**) The food intake of 8-week-old miR-124* (*n* = 5) and control (*n* = 5) mice was measured daily for five consecutive days. Statistical significance is indicated as follows: *p* < 0.05 (*), *p *< 0.01 (**), *p *< 0.0001 (****).

**Figure 2 biomolecules-15-01186-f002:**
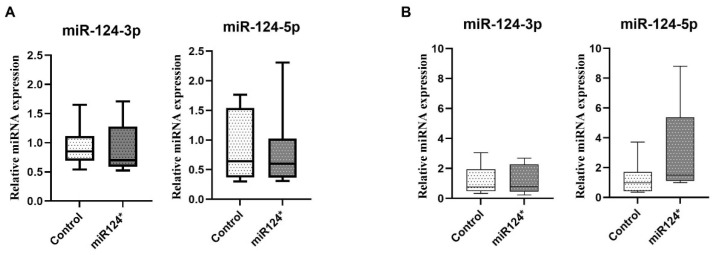
miR-124-3p and miR-124-5p miRNA expression levels on (**A**) E7.5-day-old embryos (control *n* = 19, miR-124-3p* *n* = 14) and (**B**) in adult mice hippocampus (control *n* = 7, miR-124-3p* *n* = 6).

**Figure 3 biomolecules-15-01186-f003:**
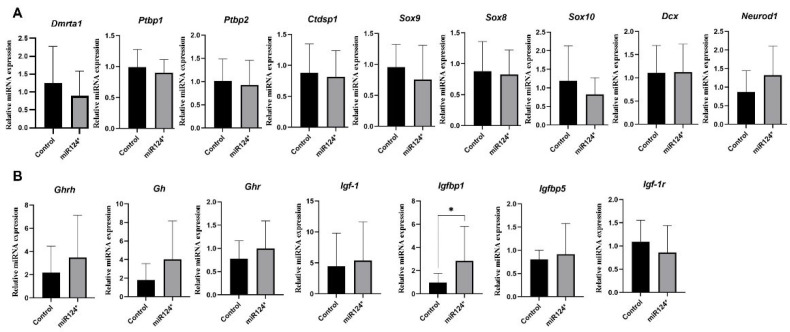
mRNA expressions of E7.5-day-old embryos; black symbol: control (*n* = 19), grey symbol: miR-124-3p* (*n* = 13) (**A**) *Dmrta1*, *Ptbp1*, *Ptbp2*, *Ctdsp1*, *Sox8*, *Sox9*, *Sox10*, *Dcx* and *Neurod1* (**B**) *Ghrh*, *Gh*, *Ghr*, *Igf-1*, *Igf1-r*, *Igfbp1*, and *Igfbp5*. *p* < 0.05 (*).

**Figure 5 biomolecules-15-01186-f005:**
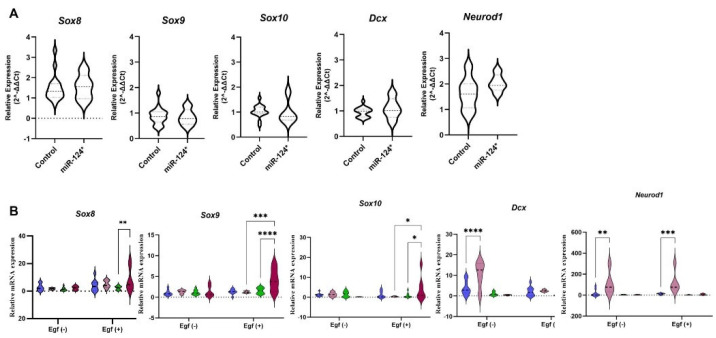
(**A**) E19.5-day-old miR-124-3p* (*n* = 5) and control (*n* = 17) fetus’ mRNA expression results in the fresh hippocampus. (**B**) mRNA expression levels of 12 and 21-day-old NSs derived from E19.5-day-old hippocampal cells: *Sox8*, *Sox9*, *Sox10*, *Dcx* and *Neurod1*. Twelve-day-old NS; blue symbols: control (*n* = 10), pink symbols: miR-124-3p* (*n* = 5), 21-day-old NS; green symbols: control (*n* = 11), red symbols: miR-124-3p* (*n* = 7). *p* < 0.05 (*), *p* < 0.01 (**), *p* < 0.001 (***), *p* < 0.0001 (****).

## Data Availability

The original contributions presented in this study are included in the article. Further inquiries can be directed to the corresponding author.

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
