# Peer review of "Dynamic Interplay Between miR-124-3p and EGF in the Regulation of Overgrowth via RNA Signaling"

_biomolecules, 2025, doi:10.3390/biom15081186_

Round 1

Reviewer 1 Report

Comments and Suggestions for Authors

Review of Biomolecules paper: July 2025

General:

The paper by Bayram et al., describes a body of work in which the investigative team sought to understand the role of miR-124-3p in regulating murine growth and cognition in a Balb/c genetic background. The approach was to microinject miR-124-3p into one cell fertilized mouse eggs and then investigate the molecular mechanisms that underlie the changes in embryonic growth and cognition. There was a focus in trying to understand the role EGF may have in the phenotypes associated with this embryonic manipulation. The work was well done, rigorous and is based, in part, on previous work published in 2009. Many of my  specific comments are meant to clarify some overstatements.

Specific Comments:

  1. Abstract, line 27- I would refrain from referring to these mice as having a ‘giant’ phenotype. Their growth is enhanced but they cannot be defined as “giant”.
  2. Abstract, line 28- I think it is an overstatement to call the 124-3p mice more cognitively active. The data supporting this finding are not robust.
  3. Conclusions, line 39- the EGF story is very tentative and correlative at best.
  4. Materials and Methods (M&M), Animals, line 75- “XXX” has been left in the text. Define the Center where the mice were housed.
  5. M&M, Animals, line 80- Need to identify “YYY”.
  6. Line 89- Change this to “Anesthesia was provided to surrogate females by….
  7. Results- The enhanced growth phenotype is not new or novel. The only new finding is that the same growth phenotype was recapitulated in the Balb/c background.
  8. Panel A- It is difficult to see any difference between the 2 pups displayed in the figure. Always best to use a blue background to highlight detail and differences. The same is true of panel B. You are displaying an albino pup on a white background. Not ideal.
  9. Line 263/264. I do not believe that there are differences in expression of miR-124-3p and miR-1245p at E7.5 and in adult hippocampus. Please restate the results or provide a statistical analysis that supports this statement.
  10. Line 275/276. The data provided does not support the statement that there was a decrease in the expression of genes described. Please reconcile.
  11. Line 291/292. The data provided does not support the statement that there was a change in expression of several of the noted genes. Please reconcile.
  12. Line 295. This paragraph needs to be reworked. The messaging is confusing, and repetitive.
  13. Line 302. Use lower case letter “a” for “In addition, At….”
  14. Line 450/451. The statement that the candidate genes noted had reduced expression is not supported by the data provided. The last sentence in the paragraph seems like hand-waving. All possible but not supported by the data provided.
  15. If possible, please refrain from using the “giant-phenotype” description. Enhanced growth or large size would be better.

Reviewer 2 Report

Comments and Suggestions for Authors

Comments about the manuscript:

“Dynamic Interplay Between miR-124-3p and EGF in Regulating the 'Giant' Phenotype by RNA Signalling Mechanism”

Epigenetic mechanisms, RNAs, and microRNAs are involved in early embryonic development. It has been shown that the microRNA miR-124-3p injected into fertilized mouse eggs (miR-124-3p*) exhibits a significantly "giant" phenotype. Various behavioral tests showed that the cognitive activity of these mice was greater than that of controls. The aim of the work presented here was to understand the still unknown molecular mechanisms that lead to these changes. To do this, the authors used young embryos and neurospheres, which exhibit the same growth, differentiation, and self-renewal properties as early embryos. They examined the mRNA expression levels of several genes in total embryos at E7.5, fetal hippocampal cells at E19.5 days, and neurospheres at 12 and 21 days. The results obtained indicate that microinjection of miR-124-3p into the fertilized egg could activate EGF, which could suppress miR-124-3p expression, leading to early neurogenesis. The mice would then exhibit active and "giant" cognitive phenotypes.

This work provides interesting and even exciting results regarding the effects of epigenetic factors and microRNAs on the development of nervous tissue, with a consequent increase in cognitive abilities. This study could be published after some refinements of the manuscript. Here are some remarks.

Page 2, line 75. What does “‘XXX’Center facilities” mean? Please specify or complete.

Page 2, line 80. What does “by the ‘YYY’.” mean? Please specify or complete  Specify or complete.

Page 3, lines 107-108. “cDNA was synthesised from obtained RNA samples using EvoScript Universal cDNA Master kit”: briefly describe the method used.

Page 3, lines 108-109. “The PreAMP Master Kit (Cat. No. 07 912 439 001, Roche, Germany) was used for preamplification”: same: briefly describe the method used.

Page 5, line 192: add a scale bar to figure 1A.

Page 6, lines 236-237 ? “Usually, the cells in women's olfactory bulbs are 43% more than in men.”: Are they women (human) or females (animal)? Men (human) or males (animal)? Specify "in humans"

Page 10, line 400. “Dcx is a neuron-expressed protein”: if it is a protein, do not use italics.

Page 10, line 407. “Dcx mRNA expression”: I think Dcx is a gene, so use italics.

Page 11, line 443. “tract-binding protein (Ptbp) expression”: Don't use italics (if it is a protein name).

Throughout the text, including figures, check whether gene names are written in italics and whether proteins are written in standard script, and correct if necessary.

Supplementary files

Figure S2, S3, S4:Add a general legend with a more explanatory title? and give more specific explanations to the images showing the results of each test.

Figure S5: add a scale bar to figures S5A.

Reviewer 3 Report

Comments and Suggestions for Authors

Previous studies showed that miR-124- 3p* mice exhibit a s 'giant' phenotype and are more cognitively active. However, the molecular mechanism remains unclear. And the epidermal growth factor (EGF) might be involved in the process. The authors therefore addressed this question. Basically the experimental design is resonable and the manuscript was organized. But the main data were only collected from PCR, it's not very strong. And the following issues need to be improved.

  1. Capitalize the first letter of the first word in the labeling of Y-axis of all figures.
  2. It'll be better if the protein expressions of all these molecules were provided. The current data obtained only by PCR are litter bit weak.
  3. The supplemental data are very important in the reviewer's view, why not put them as the formal figures?
  4. Line 66-68: Check the grammar.
  5. Line 89: Usually do not start a sentence with an Arabic number.

Round 2

Reviewer 3 Report

Comments and Suggestions for Authors

The authors replied my comments and I have no questions. The manuscript is now improved.